# Measuring the photoelectron emission delay in the molecular frame

Jonas Rist [1✉], Kim Klyssek [1], Nikolay M. Novikovskiy [2,3], Max Kircher [1], Isabel Vela-Pérez[1], Daniel Trabert [1], Sven Grundmann [1], Dimitrios Tsitsonis[1], Juliane Siebert[1], Angelina Geyer[1], Niklas Melzer [1], Christian Schwarz[1], Nils Anders[1], Leon Kaiser[1], Kilian Fehre[1], Alexander Hartung [1], Sebastian Eckart [1], Lothar Ph. H. Schmidt[1], Markus S. Schöffler[1], Vernon T. Davis[4], Joshua B. Williams[4], Florian Trinter [5,6], Reinhard Dörner [1], Philipp V. Demekhin [2✉] & Till Jahnke[1,7✉]

How long does it take to emit an electron from an atom? This question has intrigued scientists for decades. As such emission times are in the attosecond regime, the advent of attosecond metrology using ultrashort and intense lasers has re-triggered strong interest on the topic from an experimental standpoint. Here, we present an approach to measure such emission delays, which does not require attosecond light pulses, and works without the presence of superimposed infrared laser fields. We instead extract the emission delay from the interference pattern generated as the emitted photoelectron is diffracted by the parent ion's potential. Targeting core electrons in CO, we measured a 2d map of photoelectron emission delays in the molecular frame over a wide range of electron energies. The emission times depend drastically on the photoelectrons' emission directions in the molecular frame and exhibit characteristic changes along the shape resonance of the molecule.

[1] Institut für Kernphysik, J. W. Goethe-Universität, Max-von-Laue-Str. 1, 60438 Frankfurt, Germany. [2] Institut für Physik und CINSaT, Universität Kassel, Heinrich-Plett-Strasse 40, 34132 Kassel, Germany. [3] Institute of Physics, Southern Federal University, 344090 Rostov-on-Don, Russia. [4] Department of Physics, University of Nevada, Reno, NV 89557, USA. [5] FS-PETRA-S, Deutsches Elektronen-Synchrotron DESY, Notkestr. 85, 22607 Hamburg, Germany. [6] Molecular Physics, Fritz-Haber-Institut der Max-Planck-Gesellschaft, Faradayweg 4-6, 14195 Berlin, Germany. [7] European XFEL, Holzkoppel 4, 22869 Schenefeld, Germany. ✉email: rist@atom.uni-frankfurt.de; demekhin@physik.uni-kassel.de; jahnke@atom.uni-frankfurt.de

The photoelectric effect is one of the most fundamental processes used for probing atoms, molecules, and condensed matter. It has been the subject of research for more than a century and most its aspects are considered well-understood. The basic question of whether the emitted electron appears in the continuum instantaneously or after a short delay has been under investigation for decades. This question, however, needed to be translated into the language of wave mechanics of quantum objects. The translation of the classical concept of a time delay into quantum mechanical wave formalism was first accomplished seventy years ago by Eisenbud and Wigner (and later Smith) for scattering processes in a series of pioneering theoretical works[1–3]. Their findings paved the way for the understanding of the concept of a possible photoemission delay. The emitted photoelectron wave is subject to a phase shift induced by the potential of the ionized atom or molecule. This phase shift, as compared to the phase of a wave emerging from a potential-free region, has been termed the *Wigner phase*. The concept of the Wigner phase is depicted in Fig. 1. Upon encountering a potential, a plane wave $\Phi_i(x,t) = e^{ikx-i\omega t}$ changes its wavelength and as a consequence, after interacting with the potential, the phase of the plane wave is shifted by a *scattering phase* $\delta$: $\Phi_f(x,t) = e^{ikx-i\omega t+\delta}$ (see Fig. 1a). The photoeffect mimics that behavior as the photoelectron emerges from within the ion's potential, adding a corresponding half-scattering phase (see Fig. 1b). In the case of molecular photoionization, the situation becomes more complex, as the potential is anisotropic. The Wigner phase acquired in such cases depends on the emission direction of the photoelectron with respect to the molecular axis (as shown in Fig. 1c). Thus, the phase shift is a particularly sensitive, purely quantum probe of even the most subtle of features of the molecular potential. The photoelectron emission time (often referred to as Wigner time delay) is given by the derivative of the electron's phase with respect to the electron's kinetic energy $\varepsilon$ and is in the attosecond regime.

In the aftermath of the initial theoretical investigations, it took several decades before photoemission delays could be addressed in experiments. Several femtosecond-laser-related techniques have been developed during the last 20 years which give experimental access to such atomic time scales. Pioneering work by Schultze et al.[4] employed an IR-laser field-streaking approach to measure these ultrashort times. A broadband higher-harmonic attosecond light pulse was used to ionize neon atoms and a superimposed, phase-locked strong IR pulse altered the photoelectron kinetic energy depending on the electron emission time with respect to the IR pulse. From these measurements, the authors concluded that Ne(2p) electrons are emitted with an additional delay of $21 \pm 5$ as (attoseconds) compared to those liberated from the Ne(2s)-shell. This first-of-a-kind work triggered strong theoretical efforts to reproduce the emission delay in the modeling of the process, which mostly yielded much shorter emission delays of only a few attoseconds (see e.g., ref. [5] for a review). It has been pointed out since then that the streaking laser field alters the electron emission substantially and needs to be incorporated in the models used to extract the actual naturally occurring Wigner delay. More recently, the subject of photoionization of Ne has been revisited in a follow-up experiment by Isinger and coworkers[6]. The findings made there suggest that the initial measurement by Schultze et al. was (in addition) contaminated by Ne-satellite states with different Wigner delays, which were finally energetically resolved in the recent investigation.

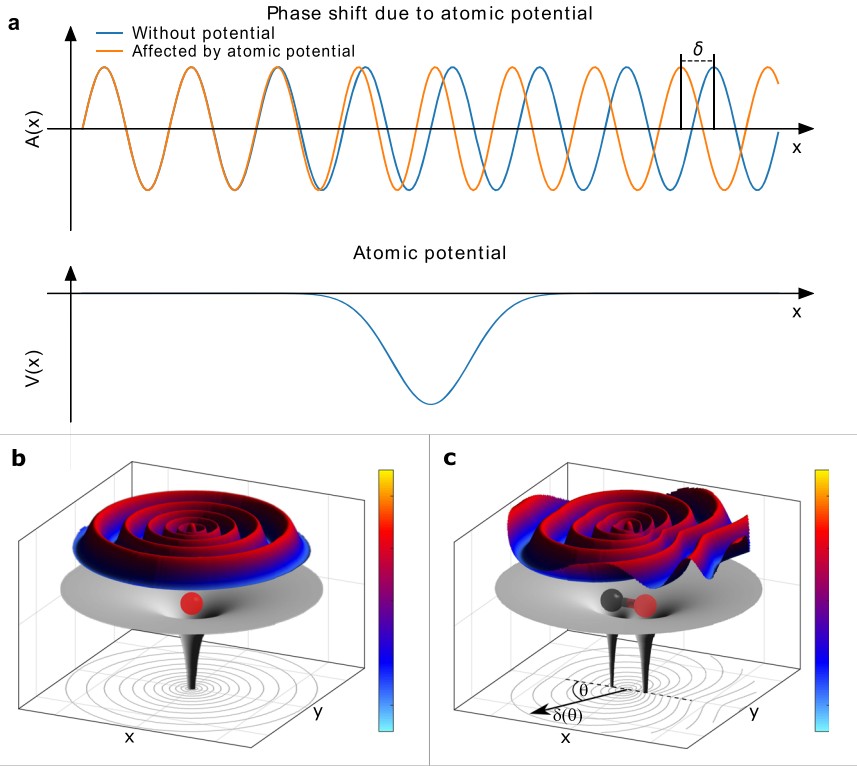

**Fig. 1 Concept of the Wigner phase. a** A plane wave with amplitude $A(x)$ passes a potential $V(x)$ from left to right. After passage, it acquires an additional phase $\delta$ due to the modulation of its wavelength by the potential. **b** Illustration of an electron wave emitted from within a single atomic potential. **c** The same as **b** but with an additional neighbouring atom resulting in a more complex molecular potential. The photoelectron wave is subject to a phase shift, which depends (due to the anisotropy of the potential) on the emission direction with respect to the molecular axis (e.g., $\delta_O \neq \delta_C$, or more generally $\delta$ depends on the emission angle with respect to the molecular axis $\theta$). The sketch depicts as an example the two-well potential of a carbon monoxide molecule.

A very detailed theoretical study by Hockett and coworkers targeted Wigner delays in small molecules[7]. Examining the emission delay in CO and $N_2$ molecules, this work provided the first fully three-dimensional Wigner delay maps in the molecular frame and showed the dependence of the emission delay on the electron kinetic energy, the molecular orientation with respect to the light polarization, and the molecular symmetry. Recently, pioneering work on molecules in the gas phase reported *stereo* Wigner delays, i.e., the difference between emission times along the direction of the carbon and the oxygen atoms of a CO molecule[8]. By using a RABBITT scheme (Reconstruction of Attosecond Beating By Interference of Two-photon Transitions)[9] and employing a COLTRIMS (Cold Target Recoil Ion Momentum Spectroscopy) reaction microscope[10], the authors resolved the phase-beating of IR-induced sidebands in the electron spectrum. From these the two-photon ionization stereo Wigner delays for the Σ- and Π-orientations of the molecule were obtained for different kinetic energy releases (i.e., ion energies) and electron energies. Further work on chiral molecules using the same techniqu showed that the Wigner delays are enantio-sensitive[11] and studies of photoionization time delays in $H_2$ showed a dependence on the dissociation process[12]. Beautiful work by Nandi and coworkers[13] demonstrated the power of the RABBITT approach by inspecting angular-integrated but vibrationally resolved Wigner delays providing deep insight into the photoionization process on a molecular shape resonance.

The experimental approaches mentioned so far rely all on attosecond light pulses for ionization and on a modification of the emission process by a superimposed strong laser pulse. Apart from being experimentally very challenging, by design these methods do not provide the Wigner delay of the one-photon ionization process, but an emission delay, which includes in addition contributions from the continuum–continuum transition due to the absorption of the IR photons. Relying on theoretical modeling the one-photon Wigner delay can be extracted from the measured two-photon process in the end. While the continuum–continuum contribution can be described rather trivially for atoms, in particular the angular dependence of this contribution can be very complex for the case of molecular photoemission[14].

In the following we present an experimental approach, which is complementary to these experimental techniques. It provides, indeed direct experimental access to the angular dependence of the Wigner delay of the one-photon molecular photoionization process. Complementary to RABBITT, however, it relies on input from theoretical modeling in order to retrieve the absolute Wigner times (i.e., our technique does not provide an angular independent offset of the delay). Using this technique we investigate core ionization of the CO molecule and extract the energy- and angle-dependant Wigner time delay for photoionization from the photoelectron's angular emission distribution in the molecular frame and compare the results with theoretical calculations.

## Results and discussion
Our approach is related to the scheme of so-called complete experiments[15,16]. An emitted electron (wave packet) $\Psi_\varepsilon$ can be written as a coherent superposition of partial waves $\Psi_\varepsilon = \sum a_{\varepsilon\ell m} Y_{\ell m}$ with angular momentum quantum number $\ell$ and magnetic quantum number $m$. Accordingly, if at a given photoelectron energy $\varepsilon$, the complex amplitudes $a_{\varepsilon\ell m}$ of each partial wave $Y_{\ell m}$ are extracted from an experiment, full information on the emitted electron and the emission process (including the angular-dependent phase) can be retrieved. Furthermore, by scanning the energy of the photons employed for the ionization, the change of the phase as a function of the electron kinetic

energy also becomes accessible. This energy derivative is exactly the emission angle $(\theta, \phi)$ and electron-energy $(\varepsilon)$-dependent Wigner delay $t_w$ for which we are looking:

$$t_w(\varepsilon, \theta, \phi) = \hbar \frac{d}{d\varepsilon}\left\{\arg\left[\Psi_\varepsilon(\theta, \phi)\right]\right\} \qquad (1)$$

A natural approach in extracting the amplitudes and phases of partial waves contributing to an emitted photoelectron signal is to examine the so-called molecular-frame photoelectron angular distributions (MFPADs). This is because the same scattering of the electron inside the molecular potential that leads to angle-dependent Wigner phases also yields a complex electron diffraction pattern, which can be recognized in the emission-direction distribution of the electron with respect to the molecular axis. Photoelectron diffraction imaging[17] relies, for example, on this effect and employs the measured diffraction pattern to gather insight into the molecular geometry[18]. In pioneering work, Cherepkov et al. have demonstrated that the extraction of amplitudes and phases from measured MFPADs is indeed possible[19] even though the fitting procedure performed there did not yield unambiguous results.

For small molecules, molecular-frame angular emission distributions can nowadays be obtained routinely using synchrotron light sources for the ionization of distinct molecular orbitals resulting in photoelectrons of a well-defined kinetic energy. The molecular-frame photoelectron angular distribution of an electron of a given kinetic energy represents the modulus squared of the coherent superposition of its partial waves:

$$\frac{d\sigma_\varepsilon}{d\Omega} \sim \left|\Psi_\varepsilon(\theta, \phi)\right|^2 \qquad (2)$$

For a linear molecule aligned along the polarization axis of the ionizing light, the problem becomes independent of the azimuthal emission angle $\phi$. As a result, only terms with $m = 0$ contribute to the sum of partial waves.

An example of such an angular distribution obtained from ionizing carbon monoxide molecules is given in Fig. 2a (see methods for details on the experiment). The red line is a fit using Eq. (5) to the measured data points employing partial waves up to $\ell = 4$. From the fitting results the phase, $\arg\left[\Psi_\varepsilon(\theta)\right]$, is calculated, which is depicted in Fig. 2b. By performing the same extraction procedure on an MFPAD obtained for an adjacent photon energy and computing numerically the energy derivative, the corresponding molecular-frame Wigner delays are obtained according to Eq. (1). The extracted Wigner delays and their dependence on the photoelectron emission angle is shown in Fig. 2c.

To confirm our experimental findings, we performed a modeling of the photoemission process using Hartree-Fock wave functions (see methods for details). The corresponding results are depicted in the middle row panels of Fig. 2 and show good agreement with the experimental findings. The two distributions shown in Fig. 2d have been computed using a relaxed-core (red line) and a frozen-core (blue line) Hartree-Fock approach. The experimental data can be considered to lie in between the two theoretical computations, slightly favoring the latter model. The measured and computed phase $\arg\left[\Psi_\varepsilon(\theta)\right]$ decreases monotonically as one moves from the direction in which the oxygen atom points towards the direction in which the carbon atom points. The molecular-frame derivative of the phase with respect to energy, i.e., the Wigner delay, exhibits more distinct features. The computed results show a sharp maximum and a deep minimum. The minimum can be clearly observed in the experimental results as well. The maximum is less pronounced but occurs in the progression towards lower photoelectron kinetic energies (see next paragraph). The Wigner delay varies by several tens of attoseconds depending on the emission angle of the

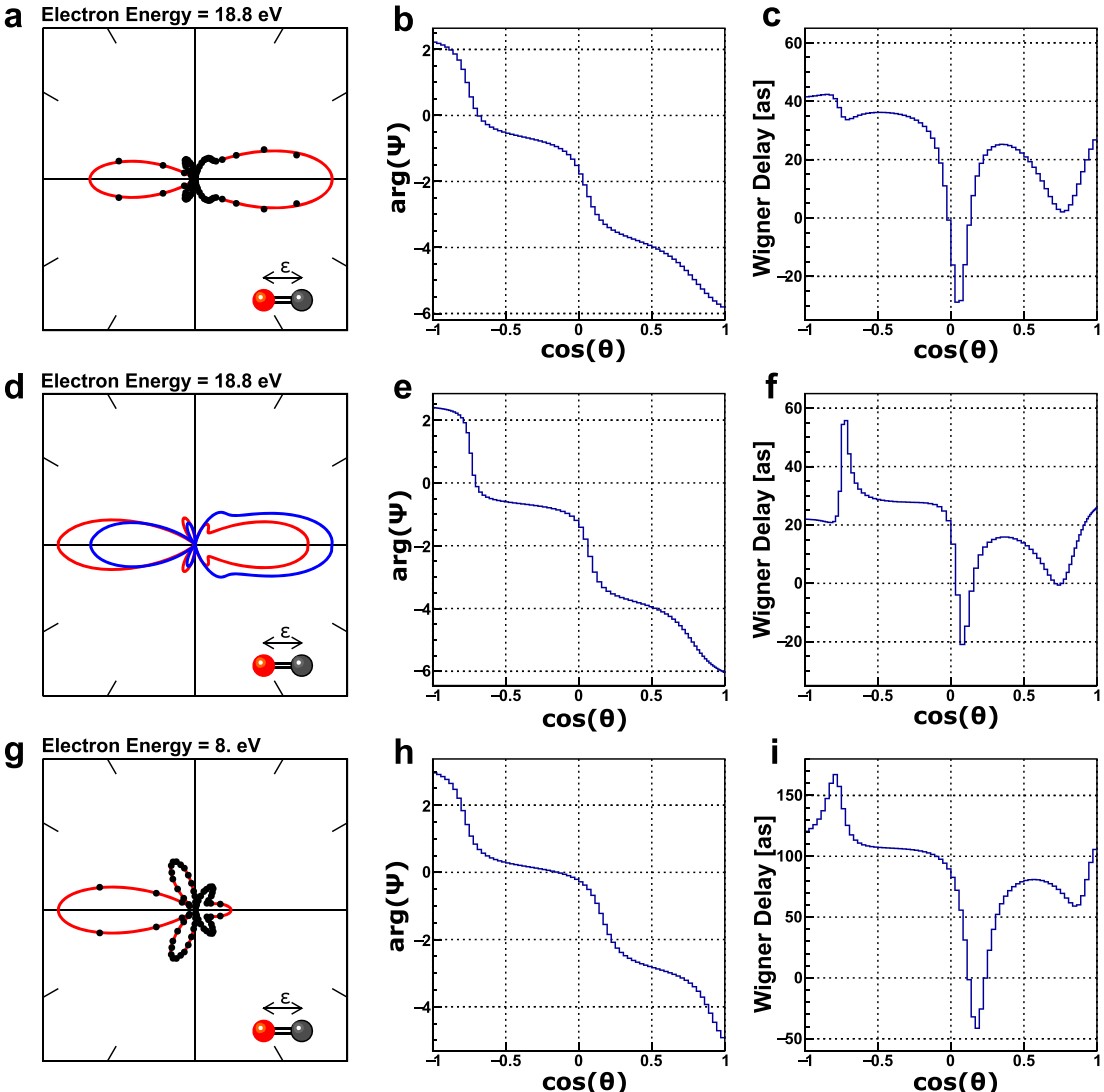

**Fig. 2 Molecular-frame photoelectron angular distribution and extracted information for $\varepsilon = 18.8$ eV and $\varepsilon = 8.0$ eV. a** Example of the emission pattern of a core electron emitted from the carbon atom of a CO molecule (black dots) for an electron energy of $\varepsilon = 18.8$ eV. The electron has been ionized by linearly-polarized synchrotron light. The molecule is oriented horizontally and parallel to the polarization axis of the photons with the emitting carbon atom pointing to the right (as depicted by the inset). The rich angular features are caused by the scattering of the emerging electron wave by the molecular potential. The statistical error bars of the data points are smaller than the markers. **b** The extracted phase $\arg\left[\Psi_\varepsilon(\theta)\right]$ and **c** molecular-frame Wigner delay. The oxygen atom is located at $\cos(\theta) = -1$, the carbon atom at $\cos(\theta) = 1$. **d–f** Corresponding results obtained from our theoretical modeling of the photoemission process. The two lines in panel **d** belong to the modeling within the relaxed-core (red) and the frozen-core (blue) Hartree-Fock approximation. The phase and the Wigner delay shown in the panels **e** and **f** correspond to the relaxed-core Hartree-Fock approximation. **g–i** Experimental results obtained for an electron energy of $\varepsilon = 8.0$ eV, i.e., recorded on the $\Sigma$-shape resonance of the molecule.

photoelectron, just as predicted by Hockett et al.[7] for the valence ionization of CO.

We have performed additional measurements of molecular-frame photoelectron angular distributions and Wigner delays in the range of the first 20 eV above the CO carbon K-threshold by scanning the photon energy from the C-K-threshold to $h\nu = 316.3$ eV. The 2D color maps shown in Fig. 3a and b depict our results of this photon energy scan as a function of electron kinetic energy and in a polar distribution of intensity, respectively. The electron energy is encoded in the radial distance from the plot's center. Figure 3a shows the molecular-frame Wigner delay map in this plotting scheme. The corresponding measured molecular-frame photoelectron angular distribution is presented in Fig. 3b. This representation of the data shows that distinct features in the Wigner delay occur at the same emission angles as minima in the

molecular-frame photoelectron angular distribution. This behavior is in line with the predictions that drastic changes in the emission delay may occur in case of the destructive interference of partial waves due to two-center or Cohen-Fano interference effects[20]. For comparison, the energy-dependent Wigner delay map resulting from our theoretical modeling and the map of the experimental data are shown in Fig. 3d and c as conventional color maps as functions of the molecular-frame photoelectron emission angle and the electron energy. The molecular axis is aligned along the direction of the light polarization ($\Sigma$-orientation). In addition, we present the corresponding histograms for molecules oriented perpendicularly to the light's polarization axis (i.e., $\Pi$-orientation) in Fig. 3e and f. The range of observed Wigner delays is much smaller for the $\Pi$-case. This is expected, as a main feature of the photoemission spectrum in the presented

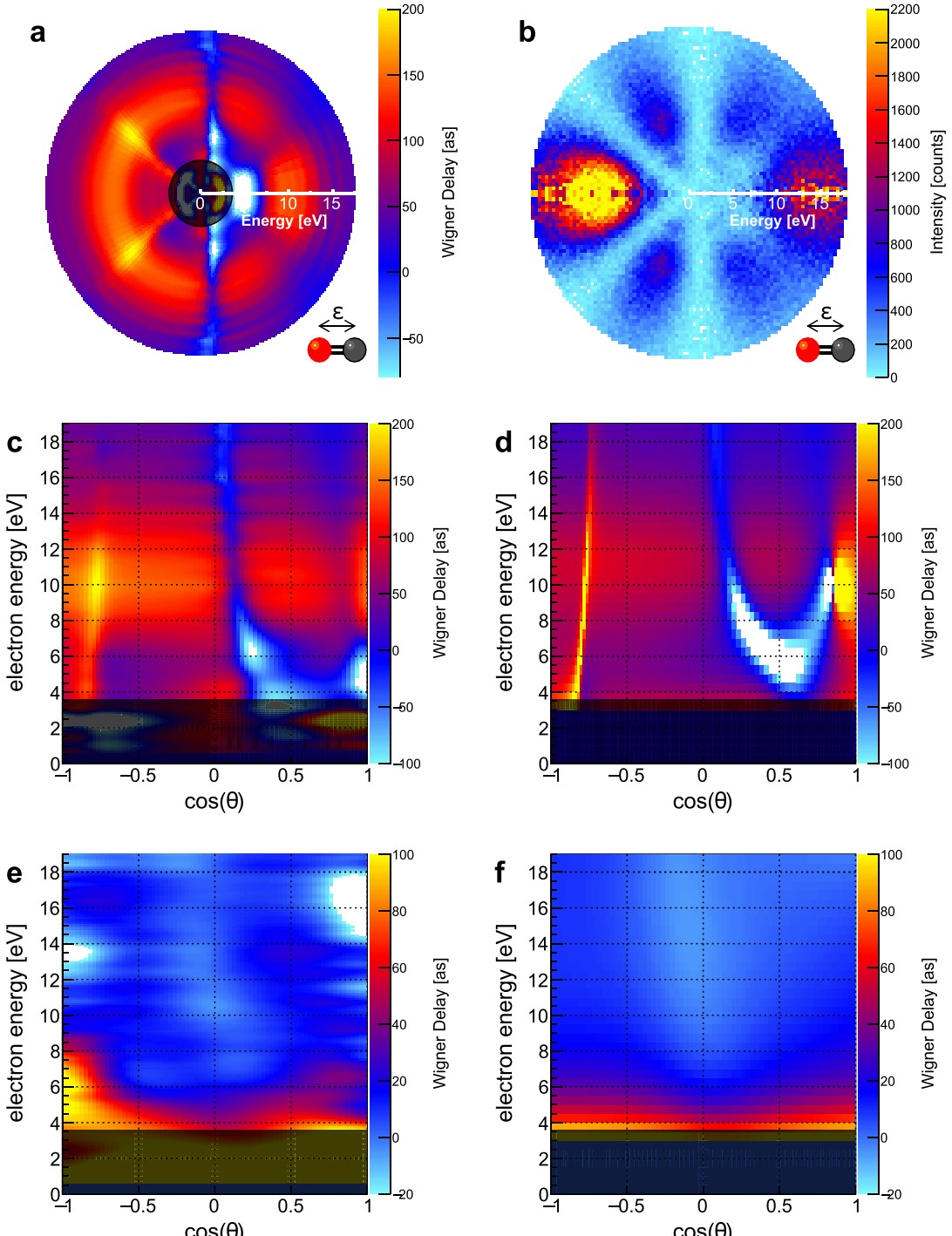

**Fig. 3 Molecular-frame photoelectron angular distributions and Wigner delay maps. a–d** The molecule is oriented along the light's polarization axis. **a** Angle-dependent Wigner delay for a range of electron kinetic energies (resulting from a scan of the photon energy). The electron energy is encoded in the distance from the center, while the value in [as] is encoded in the color scale. **b** Polar map of the molecular-frame photoelectron angular distribution in a corresponding representation. Distinct features in the Wigner delay occur under emission angles which depict minima in the angular emission distribution. **c** The same as in **a** but in a conventional color map representation. **d** Associated Wigner color map obtained from our theoretical modeling. **e**, **f** The same as **c** and **d** but for molecules oriented perpendicularly to the polarization axis of the incoming photons. For the energy region below 3.5 eV a different theoretical model would be necessary. Therefore, this region is grayed out in **a** and **c–f**.

electron-energy regime is a Σ-shape resonance, which is a broad resonance feature appearing only for the Σ-orientation with a maximum at ~8 eV[21]. Shape resonances result from a trapping of the emerging electron wave inside a centrifugal barrier, which is

present for higher-angular-momentum contributions to the emitted electron wave[22]. We clearly observe how the Wigner delay range increases as the ionization energy progresses across the resonance in the electron-energy range of $5\,\mathrm{eV} \lesssim E_e \lesssim 12\,\mathrm{eV}$

in Fig. 3c, d, while Fig. 3e, f do not depict this behavior. For direct comparison, we show the corresponding results obtained on the shape resonance in Fig. 2g–i. Furthermore, the average Wigner delay, when weighted with the angular emission probability, is ~130 as on top of the resonance, which agrees nicely with the resonance's width of ~5–6 eV. The theoretical modeling (Fig. 3d) reveals a particularly strong feature at $E_e = 9.1$ eV. The color scale is cropped in order to highlight the other details of the Wigner delay map, but the feature diverges towards negative delays and then rises abruptly to large positive delays within a very small angular range close to $\cos(\theta) = 0.8$. We do not observe a corresponding feature in the measured Wigner delay map in Fig. 3c and attribute the overestimated strength to the fixed equilibrium internuclear distance of the CO molecule employed in our theoretical modeling. We estimate that those close-lying negative and positive delays will compensate for each other if the nuclear motion is included. We notice, however, that in spite of the fixed-nuclei one-particle Hartree-Fock approximation, which is known to suffer accuracy problems where shape resonances are concerned[23], the present theory reproduces all the features observed in the experiment. The photoelectron energy region below 3.5 eV is grayed out in the histograms. It is known that in this energy region, the photoionization is strongly affected by doubly excited states[19], which are not included in our theoretical model. The angular dependence observed in the experiment, however, suggests, that these doubly excited states give rise to particularly interesting modulations in the emission time, which could be the subject of future investigations. We emphasize that in the context of our paper the good agreement between theory and experiment allows for two conclusions. First, the agreement underlines the capability of the theoretical model. Second, it validates that our technique is feasible to obtain a time domain quantity from a non-time-resolving precision experiment. The theory provides the phases of the electron's continuum wave function, thus it has direct access to the Wigner delay. The experiment infers the same quantity from measured angle-dependent amplitudes as function of photon energy.

In summary, we presented an experimental approach to extract photoelectron Wigner delays in the molecular frame. Contrary to previous approaches, measuring this attosecond-level observable does not require ultrashort laser pulses or even attosecond pulses to trigger photoemission and our approach measures the native one-photon ionization process of the unperturbed molecule without any need for dressing laser fields. Furthermore, the use of low-bandwidth synchrotron radiation allows for addressing electrons in a wide range of binding energies and distinct orbitals even in larger molecules. For more complex molecules, which are not cylindrically symmetric, complete 3D Wigner delay maps can be extracted. In the future, we will address excitation energy ranges, which are dominated by electron correlation effects (as for example those of the aforementioned doubly excited states) in order to explore the role of correlation in the Wigner phase. It has been recently demonstrated that MFPADs can be measured with X-ray free-electron lasers using the same multi-particle coincidence approach employed in the present experiment[24,25]. These measurements suggest that MFPADs (and thus 3D Wigner delay maps) from more complex molecules and processes can be obtained using X-ray free-electron laser sources and more importantly, that the temporal evolution of the MFPADs following a photoreaction will become accessible in the near future.

## Methods

**Theoretical methods.** The total amplitude for the emission of a photoelectron with energy $\varepsilon$ in the direction $(\theta, \phi)$ with respect to the axis of a diatomic molecule, which forms the Euler angle $\beta$ with the direction of linear polarization of the ionizing radiation, reads:

$$T(\varepsilon, \beta, \theta, \phi) = \sum_{\ell m k} (-i)^\ell \, \mathcal{D}^1_{k0}(\beta) \, A_{\varepsilon \ell m k} \, Y_{\ell m}(\theta, \phi). \quad (3)$$

Here, $\mathcal{D}^1_{k0}(\beta)$ are the rotation matrices (the remaining orientation Euler angles other than $\beta$ are irrelevant) and $A_{\varepsilon \ell m k} = \langle \Psi_{\varepsilon \ell m} | d^1_k | \Psi_0 \rangle$ are the dipole transition amplitudes for the emission of the partial photoelectron waves[26] with angular momentum quantum numbers $\ell$ and $m$ via the absorption of a photon of polarization $k$, all together given in the frame of a molecule. The amplitudes $A_{\varepsilon \ell m k}$ were computed by the stationary Single Center (SC) method and code[27–29], which provides an accurate theoretical description of the angle-resolved photoemission spectra. The calculations were performed in the frozen-core and relaxed-core Hartree-Fock approximations. The SC expansion of all occupied orbitals of CO was restricted to partial harmonics of $\ell_c \leq 99$, and for the photoelectron in the continuum, to $\ell_e \leq 49$. The total transition amplitude in Eq. (3) provides access to the MFPADs and Wigner delays via:

$$\frac{d\sigma}{d\Omega}(\varepsilon, \beta, \theta, \phi) = |T(\varepsilon, \beta, \theta, \phi)|^2$$
$$\text{and} \quad t_w(\varepsilon, \beta, \theta, \phi) = \hbar \frac{d}{d\varepsilon} \left\{ \arg\left[ T(\varepsilon, \beta, \theta, \phi) \right] \right\}. \quad (4)$$

The latter derivative was evaluated numerically using energy steps of 100 meV.

**Experimental methods.** The experiment was performed at beamline U49/2-PGM-1 of the BESSY II synchrotron[30]. We employed a COLTRIMS reaction microscope[10] in order to measure the momenta of the photoelectrons and the $C^+$ and $O^+$ ion pairs generated after K-shell ionization and subsequent Auger decay in coincidence. In brief, in the COLTRIMS apparatus, a supersonic gas jet of CO molecules is intersected with the synchrotron photon beam. Charged particles created within the interaction volume by photoionization are guided by homogeneous electric and magnetic fields to two time- and position-sensitive microchannel plate detectors with delay-line position readouts[31]. In this experiment, the ion arm consisted of a 5 cm long acceleration region. The electron arm of the COLTRIMS analyzer incorporated a Wiley-McLaren time-focusing scheme[32] with 6 cm acceleration region followed by a 12 cm field-free drift region. The electric field was set to 13 V/cm. A superimposed homogeneous magnetic field of 4.3 Gauss confined electrons up to a kinetic energy of 22 eV within the spectrometer volume. By measuring the positions of impact and the flight times of all particles in coincidence, the initial momentum vectors are deduced. The molecular-frame photoelectron angular distributions are then obtained by measuring the emission direction of the $C^+$ and $O^+$ ions, which are generated in a Coulomb explosion after the photoionization process and the subsequent Auger decay. It is known that the ions fragment along the initial molecular axis for an ion kinetic energy release larger than 10.2 eV[33]. We measured the photoelectron momenta in coincidence and, in that way, obtained the relative emission angle, i.e., the emission angle in the molecular frame. We scanned the photon energy in a range between 295.3 and 316.3 eV, which creates photoelectrons with a kinetic energy between 0 and 20 eV. The scanning was performed in steps of 50 meV, and the beamline exit slit was set to 150 μm corresponding to a photon energy resolution of ~150 meV.

**Data analysis.** In the case that the CO molecules are oriented along the light polarization vector, a considerable simplification of Eq. (3) occurs. In particular, the orientation angle $\beta = 0$, and the summation over the polarization index reduces to the value $k = 0$ (with the respective rotation matrices $\mathcal{D}^1_{00}(0) = 1$ and $\mathcal{D}^1_{\pm 10}(0) = 0$). As a consequence, only $\sigma$-partial waves with $m = 0$ contribute to the C 1s-photoionization channel. Thus, the respective MFPADs can be approximated as:

$$\frac{d\sigma}{d\Omega}(\varepsilon, \theta) = \left| \sum_\ell a_{\varepsilon \ell} Y_{\ell 0}(\theta) \right|^2 \quad \text{with} \quad a_{\varepsilon \ell} = (-i)^\ell A_{\varepsilon \ell 00}. \quad (5)$$

In principle, the experimental MFPADs representing the $\Sigma$-channel are obtained by selecting the subset of molecules that are aligned in parallel to the light polarization axis from the whole dataset recorded for randomly oriented molecules. However, in order to maximize the statistics of the measured dataset we adopted in part the so-called F-function formalism[34]. The electron angular distribution $I(\beta, \theta, \phi)$ after photoionization can be fully described in terms of the following $F_{LN}$ functions by:

$$I(\beta, \theta, \phi) = F_{00}(\theta) + F_{20}(\theta) P^0_2(\cos(\beta))$$
$$+ F_{21}(\theta) P^1_2(\cos(\beta)) \cos(\phi) \quad (6)$$
$$+ F_{22}(\theta) P^2_2(\cos(\beta)) \cos(2\phi)$$

where, again, $\beta$ is the angle between the molecular axis and the polarization axis and $\theta$ and $\phi$ are the polar and azimuthal angles of the outgoing photoelectron. Setting $\beta = 0$ while integrating over $\phi$ due to the rotational symmetry of the process is equivalent to selecting only the $\Sigma$-orientation. The corresponding $\theta$-dependant

distribution is given as:

$$I_\sigma(\theta) = \int_0^{2\pi} 2 \cdot I(0,\theta,\phi)\, d\phi = 4\pi(F_{00}(\theta) + F_{20}(\theta)) \qquad (7)$$

For an isotropic ionization probability, the same angular distribution is obtained by integrating over different values of $\beta$ and combining them in the following way:

$$I_\sigma(\theta) = 4A(\theta) + (4 - 3\sqrt{3})B(\theta) \qquad (8)$$

Where $A(\theta)$ and $B(\theta)$ are defined as:

$$
\begin{aligned}
A(\theta) &:= \int_{-1}^{-\frac{1}{\sqrt{3}}} I(x,\theta)dx + \int_{\frac{1}{\sqrt{3}}}^{1} I(x,\theta)dx \\
B(\theta) &:= \int_{-\frac{1}{\sqrt{3}}}^{\frac{1}{\sqrt{3}}} I(x,\theta)dx
\end{aligned}
\qquad (9)
$$

with the substitution $x = \cos(\beta)$ and $I(x,\theta) = \int_0^{2\pi} I(x,\theta,\phi)\, d\phi$.

The advantage is obvious for the experiment because now all recorded events contribute to the MFPAD of the $\Sigma$-state (instead of only a small fraction) which leads to significantly better statistics.

A similar procedure can be followed for molecules oriented perpendicularly to the light polarization ($\beta = \pi/2$) if only that part of the MFPAD distribution is considered, which lies within the plane defined by the molecular axis and the polarization axis of the light ($\phi = 0$). In this case only $\pi$-partial waves with $m = \pm 1$ emitted via absorption of photons with polarization $k = \pm 1$ contribute to the ionization (the respective rotation matrices are equal to $\mathcal{D}_{00}^1\left(\frac{\pi}{2}\right) = 0$ and $\mathcal{D}_{\pm 10}^1\left(\frac{\pi}{2}\right) = \mp\frac{1}{\sqrt{2}}$). Thereby, Eq. (3) simplifies to:

$$\frac{d\sigma}{d\Omega}(\varepsilon,\theta) = \left| -\sqrt{2} \sum_\ell a_{\varepsilon\ell} Y_{\ell 1}(\theta) \right|^2 \qquad (10)$$
$$\text{with} \quad a_{\varepsilon\ell} = (-i)^\ell\, A_{\varepsilon\ell 11}.$$

Where we used $Y_{l1} = -Y_{l(-1)}$ for $\phi = 0$ and $A_{\varepsilon\ell 11} = A_{\varepsilon\ell(-1)(-1)}$.

As before, the F-function formalism can be used to enhance the statistics for the case of $\Pi$-orientation by setting $\beta = \pi/2$ and $\phi = 0$:

$$
\begin{aligned}
I_\pi(\theta) = I(\pi/2,\theta,0) &= F_{00}(\theta) - \frac{1}{2}F_{20}(\theta) + 3F_{22}(\theta) \\
&= \frac{1}{4\pi}X + \frac{1}{8\pi}Y + \frac{3}{16}Z
\end{aligned}
\qquad (11)
$$

Where $X(\theta)$, $Y(\theta)$, and $Z(\theta)$ are defined as:

$$
\begin{aligned}
X(\theta) &:= A(\theta) + B(\theta) \\
Y(\theta) &:= 3A(\theta) + \left(3 - 3\sqrt{3}\right)B(\theta) \\
Z(\theta) &:= 8 \int_0^{\frac{1}{4}\pi} I(\theta,\phi)d\phi
\end{aligned}
\qquad (12)
$$

with $x = \cos(\beta)$ and $I(\theta,\phi) = \int_{-1}^{1} I(x,\theta,\phi)dx$.

We used the Minuit2 package of the ROOT data-analysis framework[35] to fit Eq. (5) and Eq. (10) to our MFPADs obtained by the F-function formalism in steps of 100 meV in the electron energy. For every fit the events within a range of ±1 eV were integrated. The experimentally-derived distributions are reproduced adequately when restricting the summation to spherical harmonics of $\ell \leq 4$ (as has already been previously demonstrated[36]). The main challenge of such multiparameter fitting of the real-valued MFPAD in Eq. (5) and Eq. (10) with the complex amplitudes $a_{\varepsilon\ell}$ is a lack of uniqueness. In particular, all complex amplitudes can be extracted up to one common global phase, which is, in turn, energy dependent. As a consequence, the fitting procedure can yield random jumps of this global phase as a function of energy. We employed the following solution to this problem. First, for each energy, the isotropic contribution to the total amplitude $a_{\varepsilon 0} Y_{00}$ was set to be real. As a result, all fitted amplitudes were determined up to the unknown energy-dependent global phase $\delta_0(\varepsilon)$ of the amplitude $a_{\varepsilon 0}$. Second, our theoretical calculations showed that the phase $\arg\left[\Psi_\varepsilon(\theta)\right]$, as a function of the emission angle $\theta$, depicted a monotone behavior. To fit the first dataset (of lowest electron energy), we therefore initialized the fitting algorithm with random parameters and selected a result that fulfilled the monotonicity condition. Using the fitted result obtained in that step as an input to the fitting of the next adjacent energy step, we obtained results that are consistent with our theoretical calculations. This procedure yielded reliable and stable results for many different sets of random initial parameters. We obtained the Wigner delays by using adjacent energy steps to numerically evaluate the derivative (given here, for example, for $\Sigma$-orientation):

$$t_w^\sigma(\varepsilon,\theta) = \hbar \frac{\arg\left[\sum_\ell a_{\varepsilon\ell} Y_{\ell 0}(\theta)\right] - \arg\left[\sum_\ell a_{\varepsilon'\ell} Y_{\ell 0}(\theta)\right]}{\varepsilon - \varepsilon'} \qquad (13)$$

and found that an energy step of $\varepsilon - \varepsilon' = 1$ eV yielded the most stable delays while still reproducing the details of the theoretical predictions. It should be stressed, however, that Eq. (13) yields experimental Wigner delays on a relative scale, i.e., up

to an unknown (but isotropic) energy-dependent delay, $t_w^0(\varepsilon) = \hbar \frac{d\delta_0(\varepsilon)}{d\varepsilon}$, provided by the amplitude $a_{\varepsilon 0}$. This missing isotropic delay cannot be determined from the experiment but can be fixed by calibration to the theory. Here, for each photoelectron energy, we set the angle-averaged experimental Wigner delay to the respective theoretical value.

## Data availability

The data that support the findings of this study are available in Zenodo with the identifier https://doi.org/10.5281/zenodo.556967[37].

## Code availability
The code that supports the findings of this study is available from the corresponding author upon reasonable request.

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

## Acknowledgements

The experimental work was supported by the Deutsche Forschungsgemeinschaft (DFG) and the Bundesministerium für Bildung und Forschung (BMBF). S.E. and D.T. were supported by DFG Priority Programme "Quantum Dynamics in Tailored Intense Fields" (Project No. DO 604/29-1). A.H. and K.F. are grateful for the support of the Studienstiftung des deutschen Volkes. T.J., S.G., R.D., and M.S.S. acknowledge support from the DFG via Sonderforschungsbereich 1319 (ELCH). The theoretical work was supported by the DFG Project No. DE 2366/1-2. We are very thankful for the outstanding support provided by the BESSY II staff as part of the Helmholtz Zentrum Berlin (HZB), in particular that of Ronny Golnak. T.J. is very grateful for initial discussion on the topic with Danielle Dowek. R.D. and T.J. thank Nikolai Cherepkov and Ricardo Díez Muiño for initial education on fitting partial waves many years ago.

## Author contributions

J.R., K.K., M.K., I.V.-P., D.T., S.G., D.T., J.S., A.G., N.M., C.S., N.A., L.K., K.F., A.H., S.E., L.Ph.S., M.S.S., F.T., R.D., and T.J. prepared and performed the experiment. V.T.D. and J.B.W. performed the experiment. K.K., J.R., and T.J. performed the data analysis and fitting procedure. P.V.D. and N.M.N. performed the theoretical modeling. All authors contributed to the manuscript.

## Funding

## Competing interests

The authors declare no competing interests.
