## [Peer Review File · Nature Communications]

Measuring the Photoelectron Emission Delay in the Molecular FrameReviewers' comments:

Reviewer #1 (Remarks to the Author):

The paper “Measuring the Photoelectron Emission Delay in the Molecular Frame”, by J.Rist et al, focuses on a method to derive photoionization emission time delays which does not imply the use of a source capable to reach the attosecond timescale, but it is complementary to the RABBITT and similar approaches.

Although the paper is well written and the presented results are interesting and certainly deserve publication, I am not convinced that the method described is really complementary to attosecond techniques to investigate photoionization time delays. What it does is basically to describe the angle-dependent variation in the interference between resonant and nonresonant contributions.

I am under the impression that the RABBITT experiments give a considerably deeper insight into the photoemission time delay. As an example, a previous literature article titled “Attosecond timing of electron emission from a molecular shape resonance”, by Nandi et al, *Sci. Adv.* 2020; 6 : eaba7762, is not quoted in the present work. It should be interesting to compare the results obtained by attosecond interferometric techniques with the present ones. In particular, in the literature paper the authors can follow the time delay in much more detail on going across the shape resonance, and even connect it with subtle variations in the nuclear motion. There are obvious differences, such as the investigated molecule is N₂ in that case and CO in the present one, and the photon energy range is different, but the shape resonance behavior should be illustrated at a similar depth for the two methodologies to be compared. I do not think that the present approach is complementary in the sense that it can provide different but as deep physical information.

In conclusion, I suggest the authors to transfer the paper to *Scientific Reports*, or to submit it to another more specialized journal, but I do not think that the scientific case is novel enough or insightful enough to deserve publication on *Nature Communications*.

Reviewer #2 (Remarks to the Author):

The authors have presented a nicely done frequency-resolved study of scattering phase shifts extracted from MFPADs as a function of photon energy. The data are very nice. The retrieval of the energy-dependence of the phases is also nice but is hardly discussed here.

This paper, however, contains no new physics and should not be published in a Nature journal. It should be presented for what it is - a nicely done energy-resolved MFPADs experiment - in a specialist journal on AMO Physics.

As these authors well understand, "phase shift" is the frequency domain term for "time delay", whether in scattering theory or elsewhere. This is discussed in many places, for example, the classic book by Rao and Fano "Atomic Collisions and Spectra". One can adopt either frequency domain or time domain language. But it should certainly not be presented as something "new". Very similar work in the same CO molecule, but at lower resolution and for different channels, was done around 2000 by Yagashita and by Adachi. See below.

The paper under review is not a time-resolved experiment. It is a nicely done frequency-resolved experiment of the class of experiments that have been done for many years. With the tuning of the photon energy, the authors now present this in "time domain language". Time delays in photoionization are indeed currently topical: all the more reasons to present these spectral data in a clear manner, without obfuscation of the fact that this is an energy-resolved and not time-resolved experiment.

It may be true that these authors have now varied the photon energy and therefore fitted phase shifts as a function of energy. This allows then to then calculate the Wigner-Smith delays by taking derivatives with respect to energy. But whether or not this has been done for CO, this is very well known physics and not new. Again, this is not a time-resolved experiment and it is misleading to present it as such.

These types of experiments are well known. For example, on the identical molecule CO, MFPADs were measured and discussed in:

Motoki, S., J. Adachi, Y. Hikosaka, K. Ito, M. Sano, K. Soejima, A. Yagishita, G. Raseev, and N. A. Cherepkov. "K-Shell Photoionization of CO: I. Angular Distributions of Photoelectrons from Fixed-in-Space Molecules." *Journal of Physics B: Atomic, Molecular and Optical Physics* 33, no. 20 (September 2000): 4193–4212. <https://doi.org/10.1088/0953-4075/33/20/301>.

Cherepkov, Nikolai A, G Raseev, Jun-ichi Adachi, Y Hikosaka, K Ito, Souhei Motoki, M Sano, K Soejima, and A Yagishita. "K-Shell Photoionization of CO: II. Determination of Dipole Matrix Elements and Phase

Differences." *Journal of Physics B: Atomic, Molecular and Optical Physics* 33, no. 20 (October 28, 2000): 4213–36. <https://doi.org/10.1088/0953-4075/33/20/302>.

Yagishita, Akira, Kouichi Hosaka, and Jun-Ichi Adachi. "Photoelectron Angular Distributions from Fixed-in-Space Molecules." *Journal of Electron Spectroscopy and Related Phenomena* 142, no. 3 (March 2005): 295–312. <https://doi.org/10.1016/j.elspec.2004.09.005>.

Jahnke, T., J. Titze, L. Foucar, R. Wallauer, T. Osipov, E. P. Benis, O. Jagutzki, et al. "Carbon K-Shell Photoionization of CO: Molecular Frame Angular Distributions of Normal and Conjugate Shakeup Satellites." *Journal of Electron Spectroscopy and Related Phenomena, Electron Spectroscopy Kai Siegbahn Memorial Volume*, 183, no. 1 (January 1, 2011): 48–52. <https://doi.org/10.1016/j.elspec.2010.04.010>.

The Yagashita and Adachi papers above both have CO energy-dependent partial-wave phases, arrived at in the same manner as the current manuscript, albeit perhaps at lower energy resolution and for different channels. These authors could equally well have 'calculated' Wigner-Smith delays from their data - but there is no need, as it doesn't add any new information.

I encourage the authors to present their nice energy-resolved data and analysis for what it is and to resubmit to a specialist journal in AMO Physics.

Reviewer #3 (Remarks to the Author):

The submission "Measuring the Photoelectron Emission Delay in the Molecular Frame" by Rist et al. describes a novel experimental approach to measure photoelectron emission delays (Wigner delays) from molecular systems. Following a very well-written and understandable introduction, the authors describe their new approach, which utilises a COLTRIMS-type reaction microscope to record full molecular-frame photoelectron angular distributions. These MFPADs encode the photoemission delay within the photoelectron interference pattern recorded. In particular the authors investigate the photoemission delay in CO following ionisation in the range of the Carbon K-edge. Of particular interest

is the observation of a shape resonance that clearly manifests itself in the recorded Wigner delays. The data is adequately modelled using HF methods.

The presented manuscript describes a new experimental approach to measure photoemission delays for molecular systems, which to my knowledge has not been previously demonstrated. The topic of electron dynamics during photoemission is of considerable interest across the wider physics community at the moment. The presented new methodology has several advantages over established methods, one of the key ones being that it does not rely on streaking techniques and attosecond light sources. This means that one can avoid the inherent large bandwidth of attosecond pulses and hence be much more selective with respect to the ionisation/excitation process. Using narrow-bandwidth sources is particularly important as one starts to investigate more complex molecular systems, as they still allow one to address specific resonances/states. I believe this is a novel approach beyond the current state-of-the-art that overcomes several of the technical and scientific challenges of attosecond streaking techniques that are currently employed to study photoemission delays. It will be very useful as the field turns towards larger molecular systems and of high interest to the community.

The presented methodology, data and analysis are convincing and the results are adequately modelled with established HF methods. Having said that, I do believe there is one major point the authors should address/clarify in the manuscript, which is the effect of the shape resonance on the observed MFPADs/Wigner delays, in particular because this is one of the major advantages of the approach:

-To me it is unclear from the manuscript at what exact photon energies data was collected and, more importantly, at what photon energies the data shown in Fig. 2 and 3 was collected.

-The following statement (p6/l119) is also unclear: "We clearly observe how the Wigner delay range increases as the ionization energy progresses across the resonance in the electron energy range of 5 eV . E e. 12 eV in Figs. 3c+3d" . As I understand it the data presented in Fig. 3 was collected at a fixed photon energy and what is shown is then the resulting kinetic energy (and angular) distribution of produced photoelectrons. This is not the same as scanning the ionisation energy (=photon energy) across the resonance. This should certainly be clarified.

-This also makes it difficult to compare the data in figures 2 and 3. Are they collected at the same photon energy, such that the data in fig. 2 (specified as $\epsilon = 18.8$ eV) is far away from the shape-resonance, and hence predominantly from a non-resonant process? To highlight the effect of the resonance the authors could consider adding a second line to the figures 2d-f showing the effect of the resonance (e.g. data at $eKE = 9$ eV from fig 3).

-the authors state that they have collected data "in the range of the first 20 eV above the CO carbon K-threshold." It would be interesting to see the effect as the photon energy is scanned across the resonance (which is only 5-6eV wide), e.g. in a plot of photon energy vs. average Wigner delay.

-Of course it would be desirable to fully disentangle resonant from non-resonant contributions. What is needed to do this (experimentally or theoretically)? Can the authors comment on this?

In conclusion, the authors present a powerful new approach to investigate Wigner time delays within the molecular frame, enabling one to investigate the electron dynamics at specific molecular references. Once the above mentioned points are clarified in the manuscript, I believe it is very suitable for publication in Nature Communications.

Reviewer #1:

The paper “Measuring the Photoelectron Emission Delay in the Molecular Frame”, by J. Rist et al, focuses on a method to derive photoionization emission time delays which does not imply the use of a source capable to reach the attosecond timescale, but it is complementary to the RABBITT and similar approaches. Although the paper is well written and the presented results are interesting and certainly deserve publication, I am not convinced that the method described is really complementary to attoseconds techniques to investigate photoionization time delays. What it does is basically to describe the angle-dependent variation in the interference between resonant and nonresonant contributions.

We are afraid, that the referee’s assessment is not fully correct here. We will comment on the complementarity of the two methods in the next section, where we describe the changes we made to the paper, as well, to make this point more obvious to our readers. The referee’s comment was indeed very helpful in order to understand, that we did not convey our message clearly enough.

With respect to the second point raised here by the referee: Our method actually provides the angular dependence of the photoelectron’s phase (and from its energy derivative the angular dependence of photoionization time delays). It is the MFPADs (prior to extracting the phases and performing the energy derivative) that can be considered so to say as “angle-dependent variation in the interference between resonant and non-resonant contributions”.

That our method does indeed yield the angular dependence of the delay is directly visible by comparison with the theory presented in the paper: the theory has direct access to the delays and these theoretically calculated delays nicely agree with the delays obtained by our technique.

I am under the impression that the RABBITT experiments give a considerably deeper insight into the photoemission time delay. As an example, a previous literature article titled “Attosecond timing of electron emission from a molecular shape resonance”, by Nandi et al, *Sci. Adv.* 2020; 6 : eaba7762, is not quoted in the present work. It should be interesting to compare the results obtained by attosecond interferometric techniques with the present ones. In particular, in the literature paper the authors can follow the time delay in much more detail on going across the shape resonance, and even connect it with subtle variations in the nuclear motion. There are obvious differences, such as the investigated molecule is N₂ in that case and CO in the present one, and the photon energy range is different, but the shape resonance behavior should be illustrated at a similar depth for the two methodologies to be compared. I do not think that the present approach is complementary in the sense that it can provide different but as deep physical information.

We are thankful for the referee to point out the work by Nandi et al., which is actually a perfect example to demonstrate the complementarity of RABBITT and our synchrotron-based approach. RABBITT measures by concept the observables of a 2-photon process. As such the obtained results are the sum of the sought after (1-photon) Wigner delay and the (unwanted) time delay of the continuum-continuum transition (CC). This CC delay-contribution is somewhat trivial for the atomic case but for the molecular case it has a substantial angular dependence (see e.g. work by A. Kheifets *J. Chem. Phys.* 147, 204303 (2017)). Therefore, for molecules, severe theoretical treatment is required in order to extract the angular dependence of the one-photon Wigner delay by a RABBITT measurement. RABBITT can access the energy dependence of the Wigner delay but requires severe theory input for obtaining the angular dependence. It is thereby less of a direct/theory-independent measurement tool than sometimes

alleged. Our technique on the contrary, measures really a one-photon transition and we obtain the angular dependence of the Wigner delay without theory input. However, we extract relative time-delays and rely on the theory to put the delays on the absolute scale (i.e. incorporating an energy-dependent but angular independent offset).

These different limitations reflect in the different physics insight gained in the paper by Nandi et al. and in our work. The work by Nandi et al. based on RABBITT gives an angular averaged Wigner delay across the shape resonance resolving the vibrational dependence, while our work shows the dramatic change of the angular distribution of the Wigner delay across the shape resonance.

We therefore actually believe, that our approach is indeed complementary to RABBITT and changed the introduction of our paper and in the description of our method in order to clarify this in a more accessible way. In addition, we added a reference to the paper by Nandi and coworkers.

“Beautiful work by Nandi and coworkers [Nandi.2020] demonstrated the power of the RABBITT approach by inspecting angular-integrated but vibrationally resolved Wigner delays providing deep insight into the photoionization process on a molecular shape resonance.

The experimental approaches mentioned so far rely all on attosecond light pulses for ionization and on a modification of the emission process by a superimposed strong laser pulse. Apart from being experimentally very challenging, by design these methods do not provide the Wigner delay of the one-photon ionization process, but an emission delay, which includes in addition contributions from the continuum-continuum transition due to the absorption of the IR photons. Relying on theoretical modeling the one-photon Wigner delay can be extracted from the measured two-photon process in the end. While the continuum-continuum contribution can be described rather trivially for atoms, in particular the angular dependence of this contribution can be very complex for the case of molecular photoemission [Serov.2017].

The experimental approach we present in this paper is complementary to these experimental techniques. It provides, indeed direct experimental access to the angular dependence of the Wigner delay of the one-photon molecular photoionization process. Complementary to RABBITT, however, it relies on input from theoretical modeling in order to retrieve the absolute Wigner times (i.e. our technique does not provide an angular independent offset of the delay) as we show below.”

Reviewer #2:

The authors have presented a nicely done frequency-resolved study of scattering phase shifts extracted from MFPADs as a function of photon energy. The data are very nice. The retrieval of the energy-dependence of the phases is also nice but is hardly discussed here. This paper, however, contains no new physics and should not be published in a Nature journal. It should be presented for what it is – a nicely done energy-resolved MFPADs experiment - in a specialist journal on AMO Physics.

As these authors well understand, "phase shift" is the frequency domain term for "time delay", whether in scattering theory or elsewhere. This is discussed in many places, for example, the classic book by Rao and Fano "Atomic Collisions and Spectra". One can adopt either frequency domain or time domain

language. But it should certainly not be presented as something "new". Very similar work in the same CO molecule, but at lower resolution and for different channels, was done around 2000 by Yagashita and by Adachi. See below.

The paper under review is not a time-resolved experiment. It is a nicely done frequency-resolved experiment of the class of experiments that have been done for many years. With the tuning of the photon energy, the authors now present this in "time domain language". Time delays in photoionization are indeed currently topical: all the more reasons to present these spectral data in a clear manner, without obfuscation of the fact that this is an energy-resolved and not time-resolved experiment.

It may be true that these authors have now varied the photon energy and therefore fitted phase shifts as a function of energy. This allows then to then calculate the Wigner-Smith delays by taking derivatives with respect to energy. But whether or not this has been done for CO, this is very well known physics and not new. Again, this is not a time-resolved experiment and it is misleading to present it as such. These types of experiments are well known. For example, on the identical molecule CO, MFPADs were measured and discussed in:

Motoki, S., J. Adachi, Y. Hikosaka, K. Ito, M. Sano, K. Soejima, A. Yagishita, G. Raseev, and N. A. Cherepkov. "K-Shell Photoionization of CO: I. Angular Distributions of Photoelectrons from Fixed-inSpace Molecules." *Journal of Physics B: Atomic, Molecular and Optical Physics* 33, no. 20 (September 2000): 4193–4212. <https://doi.org/10.1088/0953-4075/33/20/301>.

Cherepkov, Nikolai A, G Raseev, Jun-ichi Adachi, Y Hikosaka, K Ito, Souhei Motoki, M Sano, K Soejima, and A Yagishita. "K-Shell Photoionization of CO: II. Determination of Dipole Matrix Elements and Phase Differences." *Journal of Physics B: Atomic, Molecular and Optical Physics* 33, no. 20 (October 28, 2000): 4213–36. <https://doi.org/10.1088/0953-4075/33/20/302>.

Yagishita, Akira, Kouichi Hosaka, and Jun-Ichi Adachi. "Photoelectron Angular Distributions from Fixed inSpace Molecules." *Journal of Electron Spectroscopy and Related Phenomena* 142, no. 3 (March 2005): 295–312. <https://doi.org/10.1016/j.elspec.2004.09.005>.

Jahnke, T., J. Titze, L. Foucar, R. Wallauer, T. Osipov, E. P. Benis, O. Jagutzki, et al. "Carbon K-Shell Photoionization of CO: Molecular Frame Angular Distributions of Normal and Conjugate Shakeup Satellites." *Journal of Electron Spectroscopy and Related Phenomena, Electron Spectroscopy Kai Siegbahn Memorial Volume*, 183, no. 1 (January 1, 2011): 48–52. <https://doi.org/10.1016/j.elspec.2010.04.010>.

The Yagashita and Adachi papers above both have CO energy-dependent partial-wave phases, arrived at in the same manner as the current manuscript, albeit perhaps at lower energy resolution and for different channels. These authors could equally well have 'calculated' Wigner-Smith delays from their data - but there is no need, as it doesn't add any new information.

We are a little puzzled by some of these remarks. Firstly, we have, other than claimed by the referee, not stated in our manuscript, that we have performed a "time-resolved experiment". And we neither tried to present the adoption of the "frequency language to the time-domain as something new". We basically did, that, what the referee suggests: we report and summarize in the introduction the existing

literature of interest with respect to photoionization time delays and the correspondence between the energy and the time domain. We then explain how we extracted a time domain quantity from a clearly not time-resolved experiment. We added a further sentence as part of the conclusion of our paper in order to clarify this point another time:

“We emphasize that in the context of our paper the good agreement between theory and experiment allows for two conclusions. Firstly, the agreement underlines the capability of the theoretical model. Secondly, it validates that our technique is feasible to obtain a time domain quantity from a non-time resolving precision experiment. The theory provides the phases of the electron's continuum wave function, thus it has direct access to the Wigner delay. The experiment infers the same quantity from measured angle-dependent amplitudes as function of photon energy.”

We actually refer to and cite the pioneering work by Cherepkov/Yagishita in our manuscript. In this work, the amplitudes and phases have been extracted, just as in our work. We did not claim, that the extraction of these quantities is something novel (even though we solved the problem of unambiguous fitting results in our work). We do claim, however, that we pushed this type of analysis a next step further in order to access Wigner time delays. This was possible because of the unprecedented quality and statistics of our data. As pointed out by the referee “Time delays in photoionization are indeed currently topical.”, and we demonstrate how to access these delays by employing molecular frame electron angular distributions as a tool. This has, to our knowledge, not been done, so far.

We changed the corresponding section in the new version of our manuscript such, that it becomes more clear, what has been done in the pioneering work mentioned above

“In pioneering work, Cherepkov et al. have demonstrated that the extraction of amplitudes and phases from measured MFPADs is indeed possible [Cherepkov.2000] even though the fitting procedure performed there did not yield unambiguous results.”

With respect to the compiled list of papers (which includes one of our own works): It is obviously true, that MFPADs have been measured in the past, but, as explained before, to our knowledge MFPADs have not been used as a tool to extract Wigner time delays and their molecular frame angular dependence.

Reviewer #3:

The submission "Measuring the Photoelectron Emission Delay in the Molecular Frame" by Rist et al. describes a novel experimental approach to measure photoelectron emission delays (Wigner delays) from molecular systems. Following a very well-written and understandable introduction, the authors describe their new approach, which utilises a COLTRIMS-type reaction microscope to record full molecular-frame photoelectron angular distributions. These MFPADs encode the photoemission delay within the photoelectron interference pattern recorded. In particular the authors investigate the photoemission delay in CO following ionisation in the range of the Carbon K-edge. Of particular interest is the observation of a shape resonance that clearly manifests itself in the recorded Wigner delays. The data is adequately modelled using HF methods.

The presented manuscript describes a new experimental approach to measure photoemission delays for molecular systems, which to my knowledge has not been previously demonstrated. The topic of electron dynamics during photoemission is of considerable interest across the wider physics community at the moment. The presented new methodology has several advantages over established methods, one of the key ones being that it does not rely on streaking techniques and attosecond light sources. This means that one can avoid the inherent large bandwidth of attosecond pulses and hence be much more selective with respect to the ionisation/excitation process. Using narrow-bandwidth sources is particularly important as one starts to investigate more complex molecular systems, as they still allow one to address specific resonances/states. I believe this is a novel approach beyond the current state-of-the-art that overcomes several of the technical and scientific challenges of attosecond streaking techniques that are currently employed to study photoemission delays. It will be very useful as the field turns towards larger molecular systems and of high interest to the community.

The presented methodology, data and analysis are convincing and the results are adequately modelled with established HF methods. Having said that, I do believe there is one major point the authors should address/clarify in the manuscript, which is the effect of the shape resonance on the observed MFPADs/Wigner delays, in particular because this is one of the major advantages of the approach:

-To me it is unclear from the manuscript at what exact photon energies data was collected and, more importantly, at what photon energies the data shown in Fig. 2 and 3 was collected.

We thank the referee for her/his kind assessment of our work. The data were recorded by scanning the photon energy in an energy range from the C-K-threshold to approx. 20 eV above the threshold. The example shown in Fig. 2(a,b) was recorded at a photon energy of 18.8 eV above the K-threshold (i.e., $h\nu \sim 315$ eV). In order to extract the Wigner delays, the energy derivative of the extracted phase needs to be computed. So, Fig.2(c) was obtained by employing two MFPADs measured at 315 eV and a photon energy close by. We changed the corresponding parts of our manuscript to explain this more clearly:

“From the fitting results the phase, $\arg[\Psi\varepsilon(\theta)]$, is calculated, which is depicted in Fig. 2b. By performing the same extraction procedure on an MFPAD obtained for an adjacent photon energy and computing numerically the energy derivative, the corresponding molecular-frame Wigner delays are obtained according to Eq. 1. The extracted Wigner delays and their dependence on the photoelectron emission angle is shown in Fig. 2c.”

-The following statement (p6/l119) is also unclear: “We clearly observe how the Wigner delay range increases as the ionization energy progresses across the resonance in the electron energy range of 5 eV . E e. 12 eV in Figs. 3c+3d” . As I understand it the data presented in Fig. 3 was collected at a fixed photon energy and what is shown is then the resulting kinetic energy (and angular) distribution of produced photoelectrons. This is not the same as scanning the ionisation energy (=photon energy) across the resonance. This should certainly be clarified.

We are afraid, this is a misunderstanding. We performed indeed a photon energy scan from the C Kthreshold at $h\nu=296.3$ eV to approx. $h\nu\approx 316.3$ eV. We have pointed this concept out more clearly in the amended version of our manuscript:

“We have performed additional measurements of molecular-frame photoelectron angular distributions and Wigner delays in the range of the first 20 eV above the CO carbon K-threshold by scanning the photon energy from the C-K-threshold to $h\nu = 316.3$ eV.”

-This also makes it difficult to compare the data in figures 2 and 3. Are they collected at the same photon energy, such that the data in fig. 2 (specified as $\epsilon = 18.8$ eV) is far away from the shape resonance, and hence predominantly from a non-resonant process? To highlight the effect of the resonance the authors could consider adding a second line to the figures 2d-f showing the effect of the resonance (e.g. data at $eKE = 9$ eV from fig 3).

We have changed the figure in accordance to the suggestions by the referee and hope that with the changes described before, we are now able to clearly convey, that we scanned the photon energy and the energy of electrons depicted in Fig. 2 and 3 results from $E_e = h\nu - E_{\text{bind}}$. We are in particular thankful for the requested change, as we realized, that due to a typo in the macro (which we used to generate the panels of Fig. 2 from the color-maps shown in Fig. 3) the histograms shown in Fig. 2 were scaled up by a factor of two. This has now been corrected and Figures 2 and 3 are now consistent.

-the authors state that they have collected data “in the range of the first 20 eV above the CO carbon Kthreshold.” It would be interesting to see the effect as the photon energy is scanned across the resonance (which is only 5-6eV wide), e.g. in a plot of photon energy vs. average Wigner delay.

This behavior is visible in Fig. 3. As expected, the Wigner delays vary more strongly on the shape resonance.

-Of course it would be desirable to fully disentangle resonant from non-resonant contributions. What is needed to do this (experimentally or theoretically)? Can the authors comment on this?

This is indeed a very interesting question and its answer requires the development of further theoretical approaches. As the referee might be aware, we have suggested to the editor to consider our work in connection with a companion paper by Holzmeier and coworkers. This work provides as one of their findings (apart from demonstrating the viability of a similar fitting procedure as the one described in our work (which they developed independently)) the disentanglement of the non-resonant and the resonant contribution appearing on a shape resonance of NO molecules in the valence ionization regime. We therefore decided not to pursue this indeed very interesting route and focused in our work on the description of the extraction procedure, depicting angle resolved Wigner-delay-map, highlighting its generality and putting it in context to other measurement approaches known from the attosecond community.

REVIEWERS' COMMENTS

Reviewer #1 (Remarks to the Author):

I appreciate the authors' response to my criticisms and those of the other referees. As I said in my previous report, I never questioned the validity of the obtained experimental and theoretical results. I think that the complementarity of the present approach with time-resolved measurements is now made clearer. Therefore, I am inclined to consider the present work as suitable for publication on Nature Communications.

Reviewer #2 (Remarks to the Author):

I thank the authors for their detailed reply.

It seems that we are in agreement that there is not a new conceptual advance here but rather better implementation and statistics, as the authors clearly state themselves:

"We did not claim, that the extraction of these quantities is something novel (even though we solved the problem of unambiguous

fitting results in our work). We do claim, however, that we pushed this type of analysis a next step further in order to access Wigner time delays. This was possible because of the unprecedented quality and statistics of our data."

Therefore, my conclusion remains the same. This is not a new technique - it is a better implementation. This is a very nicely done energy-resolved MFPAD experiment. It is always possible to represent such data in either the frequency or time domains. As the authors agree, this too is not novel.

Given that the above, I again recommend that the authors submit this very nice work to a specialist journal in AMO Physics.

Reviewer #3 (Remarks to the Author):

The authors have adequately addressed all points raised in my previous review.

I continue to be excited about the presented new and complimentary approach to extract attosecond photoemission delays for molecular systems and recommend publication of this manuscript.